# ON THE EXPRESSIVE POWER OF DEEP NEURAL NETWORKS

**Maithra Raghu**
Google Brain and Cornell University

**Ben Poole**
Stanford University and Google Brain

**Jon Kleinberg**
Cornell University

**Surya Ganguli**
Stanford University

**Jascha Sohl-Dickstein**
Google Brain

## ABSTRACT

We study the expressive power of deep neural networks before and after training. Considering neural nets after random initialization, we show that three natural measures of expressivity all display an exponential dependence on the depth of the network. We prove, theoretically and experimentally, that all of these measures are in fact related to a fourth quantity, *trajectory length*. This quantity grows exponentially in the depth of the network, and is responsible for the depth sensitivity observed. These results translate to consequences for networks during and after training. They suggest that parameters earlier in a network have greater influence on its expressive power – in particular, given a layer, its influence on expressivity is determined by the *remaining depth* of the network after that layer. This is verified with experiments on MNIST and CIFAR-10. We also explore the effect of training on the input-output map, and find that it trades off between the stability and expressivity of the input-output map.

## 1 INTRODUCTION

Neural network architectures have proven "unreasonably effective" (LeCun, 2014; Karpathy, 2015) on many tasks, including image classification (Krizhevsky et al., 2012), identifying particles in high energy physics (Baldi et al., 2014), playing Go (Silver et al., 2016), and modeling human student learning (Piech et al., 2015). Despite their power, we have limited knowledge of how and why neural networks work, and much of this understanding is qualitative and heuristic.

To aim for a more precise understanding, we must disentangle factors influencing their effectiveness, *trainability*, or how well they can be fit to data; *generalizability*, or how well they perform on novel examples; and *expressivity*, or the set of functions they can compute.

All three of these properties are crucial for understanding the performance of neural networks. Indeed, for success at a particular task, neural nets must first be effectively trained on a dataset, which has prompted investigation into properties of objective function landscapes (Dauphin et al., 2014; Goodfellow et al., 2014; Choromanska et al., 2014), and the design of optimization procedures specifically suited to neural networks (Martens and Grosse, 2015). Trained networks must also be capable of generalizing to unseen data, and understanding generalization in neural networks is also an active line of research: (Hardt et al., 2015) bounds generalization error in terms of stochastic gradient descent steps, (Sontag, 1998; Bartlett and Maass, 2003; Bartlett et al., 1998) study generalization error through VC dimension, and (Hinton et al., 2015) looks at developing smaller models with better generalization.

In this paper, we focus on the third of these properties, *expressivity* — the capability of neural networks to accurately represent different kinds of functions. As the class of functions achievable by a neural network is dependent on properties of its architecture, e.g. depth, width, fully connected, convolutional, etc; a better understanding of expressivity may greatly inform architectural choice and inspire more tailored training methods.

Prior work on expressivity has yielded many fascinating results by directly examining the achievable functions of a particular architecture. Through this, neural networks have been shown to be

universal approximators (Hornik et al., 1989; Cybenko, 1989), and connections between boolean and threshold networks and ReLU networks developed in (Maass et al., 1994; Pan and Srikumar, 2015). The inherent expressivity due to increased depth has also been studied in (Eldan and Shamir, 2015; Telgarsky, 2015; Martens et al., 2013; Bianchini and Scarselli, 2014), and (Pascanu et al., 2013; Montufar et al., 2014), with the latter introducing the number of linear regions as a measure of expressivity.

These results, while compelling, also highlight limitations of much of the existing work on expressivity. Much of the work examining achievable functions relies on unrealistic architectural assumptions, such as layers being exponentially wide (in the universal approximation theorem). Furthermore, architectures are often compared via 'hardcoded' weight values – a *specific* function that can be represented efficiently by one architecture is shown to only be inefficiently approximated by another.

Comparing architectures in such a fashion limits the generality of the conclusions, and does not entirely address the goal of understanding expressivity — to provide characteristic properties of a typical set of networks arising from a particular architecture, and extrapolate to practical consequences.

**Random networks**    To address this, we begin our analysis of network expressivity on a family of networks arising in practice — the behaviour of networks after *random initialization*. As random initialization is the starting point to most training methods, results on random networks provide natural *baselines* to compare trained networks with, and are also useful in highlighting properties of trained networks (see Section 3). The expressivity of these random networks is largely unexplored. In previous work (Poole et al., 2016) we studied the propagation of *Riemannian curvature* through random networks by developing a mean field theory approach, which quantitatively supports the conjecture that deep networks can disentangle curved manifolds in input space. Here, we take a more direct approach, exactly relating the architectural properties of the network to measures of expressivity and exploring the consequences for trained networks.

**Measures of Expressivity**    In particular, we examine the effect of the depth and width of a network architecture on three different natural measures of functional richness: number of transitions, activation patterns, and number of dichotomies.

**Transitions:** Counting neuron transitions is introduced indirectly via linear regions in (Pascanu et al., 2013), and provides a tractable method to estimate the degree of non-linearity of the computed function.

**Activation Patterns:** Transitions of a single neuron can be extended to the outputs of all neurons in all layers, leading to the (global) definition of a network *activation pattern*, also a measure of non-linearity. Network activation patterns directly show how the network partitions input space (into *convex polytopes*), through connections to the theory of *hyperplane arrangements*.

**Dichotomies:** We also measure the *heterogeneity* of a generic class of functions from a particular architecture by counting dichotomies, 'statistically dual' to sweeping input in some cases. This measure reveals the importance of *remaining depth* in expressivity, in both simulation and practice.

**Connection to Trajectory Length**    All three measures display an exponential increase with depth, but not width (most strikingly in Figure 4). We discover and prove the underlying reason for this – all three measures are directly proportional to a fourth quantity, *trajectory length*. In Theorem 1) we show that trajectory length grows exponentially with depth (also supported by experiments, Figure 1) which explains the depth sensitivity of the other three measures.

**Consequences for Trained Networks**    Our empirical and theoretical results connecting transitions and dichotomies to trajectory length also suggest that parameters earlier in the network should have exponentially greater influence on parameters later in the network. In other words, the influence on expressivity of parameters, and thus layers, is directly related to the *remaining depth* of the network after that layer. Experiments on MNIST and CIFAR-10 support this hypothesis — training only earlier layers leads to higher accuracy than training only later layers. We also find, with experiments on MNIST, that the training process trades off between the *stability* of the input-output map and its expressivity.

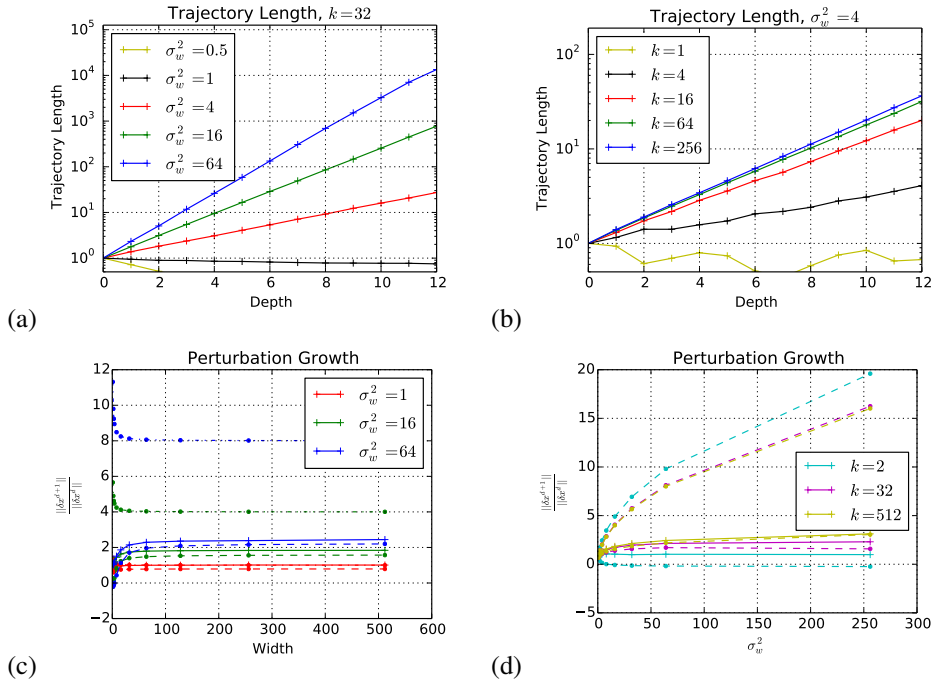

Figure 1: The exponential growth of trajectory length with depth, in a random deep network with hard-tanh nonlinearities. A circular trajectory is chosen between two random vectors. The image of that trajectory is taken at each layer of the network, and its length measured. *(a,b)* The trajectory length vs. layer, in terms of the network width $k$ and weight variance $\sigma_w^2$, both of which determine its growth rate. *(c,d)* The average ratio of a trajectory's length in layer $d+1$ relative to its length in layer $d$. The solid line shows simulated data, while the dashed lines show upper and lower bounds (Theorem 1). Growth rate is a function of layer width $k$, and weight variance $\sigma_w^2$.

## 2 GROWTH OF TRAJECTORY LENGTH AND MEASURES OF EXPRESSIVITY

In this section we examine random networks, proving and empirically verifing the exponential growth of trajectory length with depth. We then relate trajectory length to transitions, activation patterns and dichotomies, and show their exponential increase with depth.

### 2.1 NOTATION AND DEFINITIONS

Let $F_W$ denote a neural network. In this section, we consider architectures with input dimension $m$, $n$ hidden layers all of width $k$, and (for convenience) a scalar readout layer. (So, $F_W : \mathbb{R}^m \to \mathbb{R}$.) Our results mostly examine the cases where $\phi$ is a hard-tanh (Collobert and Bengio, 2004) or ReLU nonlinearity. All hard-tanh results carry over to tanh with additional technical steps.

We use $v_i^{(d)}$ to denote the $i^{th}$ neuron in hidden layer $d$. We also let $x = z^{(0)}$ be an input, $h^{(d)}$ be the hidden representation at layer $d$, and $\phi$ the non-linearity. The weights and bias are called $W^{(d)}$ and $b^{(d)}$ respectively. So we have the relations

$$h^{(d)} = W^{(d)} z^{(d)} + b^{(d)}, \qquad\qquad z^{(d+1)} = \phi(h^{(d)}). \qquad (1)$$

**Definitions** Say a neuron *transitions* when it switches linear region in its activation function (i.e. for ReLU, switching between zero and linear regimes, for hard-tanh, switching between negative saturation, unsaturated and positive saturation). For hard-tanh, we refer to a *sign transition* as the neuron switching sign, and a *saturation transition* as switching from being saturated between $\pm 1$. The *Activation Pattern* of the entire network is defined by the output regions of every neuron. More precisely, given an input $x$, we let $\mathcal{A}(F_W, x)$ be a vector representing the activation region of every hidden neuron in the network. So for a ReLU network $F_W$, we can take $\mathcal{A}(F_W, x) \in \{-1, 1\}^{nk}$ with $-1$ meaning the neuron is in the zero regime, and $1$ meaning it is in the linear regime. For

hard-tanh network $F_W$, we can (overloading notation slightly) take $\mathcal{A}(F_W, x) \in \{-1, 0, 1\}^{nk}$. The use of this notation will be clear by context. Given a set of inputs $S$, we say a *dichotomy* over $S$ is a labeling of each point in $S$ as $\pm 1$.

We assume the weights of our neural networks are initialized as random Gaussians, with appropriate variance scaling to account for width, i.e. $W_{ij}^{(d)} \sim \mathcal{N}(0, \sigma_w^2/k)$, and biases $b_i^{(d)} \sim \mathcal{N}(0, \sigma_b^2)$. In the analysis below, we sweep through a one dimensional input trajectory $x(t)$. The results hold for almost any such smooth $x(t)$, provided that at any point $x(t)$, the trajectory direction has some non-zero magnitude perpendicular to $x(t)$.

## 2.2 TRAJECTORY LENGTH AND NEURON TRANSITIONS

We first prove how the trajectory length grows, and relate it to neuron transitions.

### 2.2.1 BOUND ON TRAJECTORY LENGTH GROWTH

We prove (with a more exact lower bound in the Appendix):

**Theorem 1.** Bound on Growth of Trajectory Length *Let $F_W$ be a hard tanh random neural network and $x(t)$ a one dimensional trajectory in input space. Define $z^{(d)}(x(t)) = z^{(d)}(t)$ to be the image of the trajectory in layer $d$ of $F_W$, and let $l(z^{(d)}(t)) = \int_t \left\| \frac{dz^{(d)}(t)}{dt} \right\| dt$ be the arc length of $z^{(d)}(t)$. Then*

$$\mathbb{E}\left[ l(z^{(d)}(t)) \right] \geq O\left( \left( \frac{\sigma_w}{(\sigma_w^2 + \sigma_b^2)^{1/4}} \cdot \frac{\sqrt{k}}{\sqrt{\sqrt{\sigma_w^2 + \sigma_b^2} + k}} \right)^d \right) l(x(t))$$

This bound is *tight* in the limits of large $\sigma_w$ and $k$. An immediate Corollary for $\sigma_b = 0$, i.e. no bias, is

**Corollary 1.** Bound on Growth of Trajectory Length Without Bias *For $F_W$ with zero bias, we have*

$$\mathbb{E}\left[ l(z^{(d)}(t)) \right] \geq O\left( \left( \frac{\sqrt{\sigma_w k}}{\sqrt{\sigma_w + k}} \right)^d \right) l(x(t))$$

The theorem shows that the image of a trajectory in layer $d$ has grown *exponentially in $d$*, with the scaling $\sigma_w$ and width of the network $k$ determining the base. We additionally state and prove a simple $O(\sigma_w^d)$ growth upper bound in the Appendix. Figure 1 demonstrates this behavior in simulation, and compares against the bounds. Note also that if the variance of the bias is comparatively too large i.e. $\sigma_b >> \sigma_w$, then we no longer see exponential growth. This corresponds to the phase transition described in (Poole et al., 2016).

The proof can be found in the Appendix. A rough outline is as follows: we look at the expected growth of the difference between a point $z^{(d)}(t)$ on the curve and a small perturbation $z^{(d)}(t + dt)$, from layer $d$ to layer $d + 1$. Denoting this quantity $\left\| \delta z^{(d)}(t) \right\|$, we derive a recurrence relating $\left\| \delta z^{(d+1)}(t) \right\|$ and $\left\| \delta z^{(d)}(t) \right\|$ which can be composed to give the desired growth rate.

The analysis is complicated by the statistical dependence on the image of the input $z^{(d+1)}(t)$. So we instead form a recursion by looking at the component of the difference perpendicular to the image of the input in that layer, i.e. $\left\| \delta z_\perp^{(d+1)}(t) \right\|$. For a typical trajectory, the perpendicular component preserves a fraction $\sqrt{\frac{k-1}{k}}$ of the total trajectory length, and our derived growth rate thus provides a close lower bound, as demonstrated in Figure 1(c,d).

### 2.2.2 RELATION TO NUMBER OF TRANSITIONS

Further experiments (Figure 2) show:

**Observation 1.** The number of sign transitions in a network $F_W$ is directly proportional to the length of the latent image of the curve, $z^{(n)}(t)$.

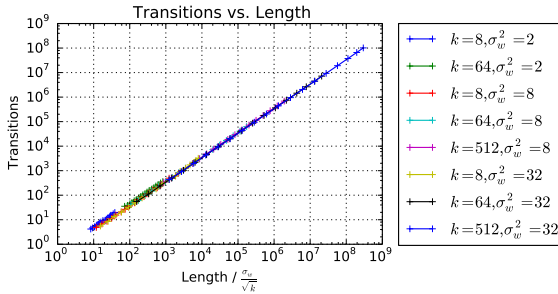

Figure 2: The number of transitions is linear in trajectory length. Here we compare the empirical number of sign changes to the length of the trajectory, for images of the same trajectory at different layers of a hard-tanh network. We repeat this comparison for a variety of network architectures, with different network width $k$ and weight variance $\sigma_w^2$.

We intuit a reason for this observation as follows: note that for a network $F_W$ with $n$ hidden layers, the linear, one dimensional, readout layer outputs a value by computing the inner product $W^{(n)}z^{(n)}$. The sign of the output is then determined by whether this quantity is $\geq 0$ or not. In particular, the decision boundary is a *hyperplane*, with equation $W^{(n)}z^{(n)} = 0$. So, the number of transitions the output neuron makes as $x(t)$ is traced is exactly the number of times $z^{(n)}(t)$ crosses the decision boundary. As $F_W$ is a random neural network, with signs of weight entries split purely randomly between $\pm 1$, it would suggest that points far enough away from each other would have independent signs, i.e. a direct proportionality between the length of $z^{(n)}(t)$ and the number of times it crosses the decision boundary.

We can also prove this in the special case when $\sigma_w$ is very large. Note that by Theorem 1, very large $\sigma_w$ results in a trajectory growth rate of

$$g(k, \sigma_w, \sigma_b, n) = O\left( \left( \frac{\sqrt{k}}{\sqrt{1 + \frac{\sigma_b^2}{\sigma_w^2}}} \right)^n \right)$$

Large $\sigma_w$ also means that for any input (bounded away from zero), almost all neurons are saturated. Furthermore, any neuron transitioning from $1$ to $-1$ (or vice versa) does so almost instantaneously. In particular, at most *one* neuron within a layer is transitioning for any input. We can then show that in the large $\sigma_w$ limit the number of transitions *matches* the trajectory length (proof in the Appendix, via a reduction to magnitudes of independent Gaussians):

**Theorem 2.** Number of transitions in large weight limit *Given $F_W$, in the very large $\sigma_w$ regime, the number of sign transitions of the network as an input $x(t)$ is swept is of the order of $g(k, \sigma_w, \sigma_b, n)$.*

## 2.3 TRANSITIONS AND ACTIVATION PATTERNS

We can generalize the 'local' notion of expressivity of a neuron's sign transitions to a 'global' measure of *activation patterns* over the entire network. We can formally relate network activation patterns to specific *hyperplane arrangements*, which allows proof of three exciting results.

First, we can precisely state the effect of a neural network on input space, also visualized in Figure 3

**Theorem 3.** Regions in Input Space *Given a network $F_W$ with with ReLU or hard-tanh activations, input space is partitioned into convex regions (polytopes), with $F_W$ corresponding to a different linear function on each region.*

This results in a *bijection* between transitions and activation patterns for 'well-behaved' trajectories, see the proof of Theorem 3 and Corollary 2 in Appendix.

Finally, returning to the goal of understanding expressivity, we can upper bound the expressive power of a particular architecture according to the activation patterns measure:

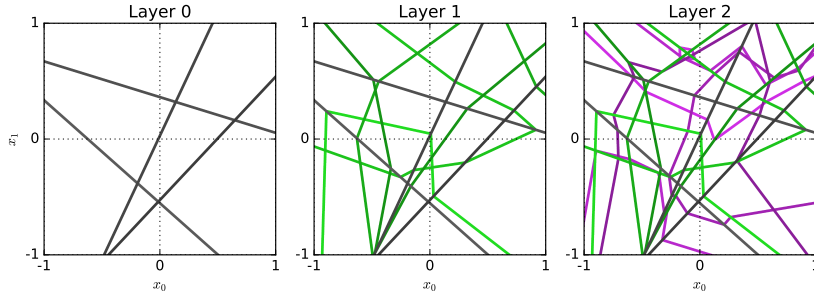

Figure 3: Deep networks with piecewise linear activations subdivide input space into convex polytopes. Here we plot the boundaries in input space separating unit activation and inactivation for all units in a three layer ReLU network, with four units in each layer. The left pane shows activation boundaries (corresponding to a hyperplane arrangement) in gray for the first layer only, partitioning the plane into regions. The center pane shows activation boundaries for the first two layers. Inside *every* first layer region, the second layer activation boundaries form a *different* hyperplane arrangement. The right pane shows activation boundaries for the first three layers, with different hyperplane arrangements inside all first and second layer regions. This final set of convex regions correspond to different activation patterns of the network – i.e. different linear functions.

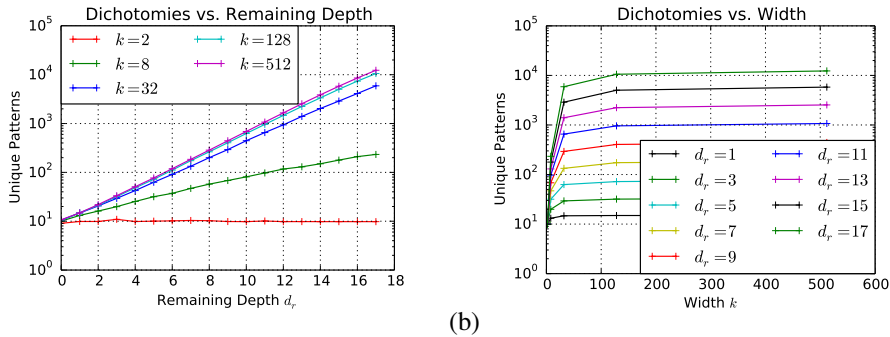

Figure 4: The number of functions achievable in a deep hard-tanh network by sweeping a single layer's weights along a one dimensional trajectory is exponential in the remaining depth, but increases only slowly with network width. Here we plot the number of classification dichotomies over $s = 15$ input vectors achieved by sweeping the first layer weights in a hard-tanh network along a one-dimensional great circle trajectory. We show this *(a)* as a function of remaining depth for several widths, and *(b)* as a function of width for several remaining depths. All networks were generated with weight variance $\sigma_w^2 = 8$, and bias variance $\sigma_b^2 = 0$.

**Theorem 4.** (Tight) Upper bound for Number of Activation Patterns *Given a neural network $F_W$, inputs in $\mathbb{R}^m$, with ReLU or hard-tanh activations, and with $n$ hidden layers of width $k$, the number of activation patterns grows at most like $O(k^{mn})$ for ReLU, or $O((2k)^{mn})$ for hard-tanh.*

## 2.4 DICHOTOMIES: A NATURAL DUAL

So far, we have looked at the effects of depth and width on the expressiveness (measured through transitions and activations) of a generic function computed by that network architecture. These measures are directly related to trajectory length, which is the underlying reason for exponential depth dependence.

A natural extension is to study a *class* of functions that might arise from a particular architecture. One such class of functions is formed by sweeping the *weights* of a network instead of the input. More formally, we pick random *matrices*, $W, W'$, and consider the weight interpolation $W \cos(t) + W' \sin(t)$, each choice of weights giving a different function. When this process is applied to just the first layer, we have a *statistical duality* with sweeping a circular input.

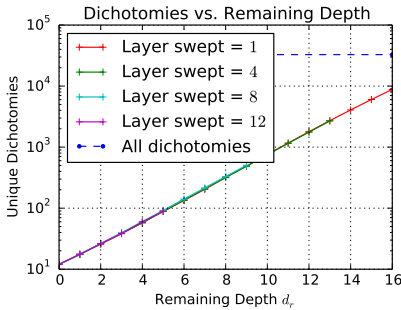

Figure 5: Expressive power depends only on remaining network depth. Here we plot the number of dichotomies achieved by sweeping the weights in different network layers through a 1-dimensional great circle trajectory, as a function of the remaining network depth. The number of achievable dichotomies does not depend on the total network depth, only on the number of layers above the layer swept. All networks had width $k = 128$, weight variance $\sigma_w^2 = 8$, number of datapoints $s = 15$, and hard-tanh nonlinearities. The blue dashed line indicates all $2^s$ possible dichotomies for this random dataset.

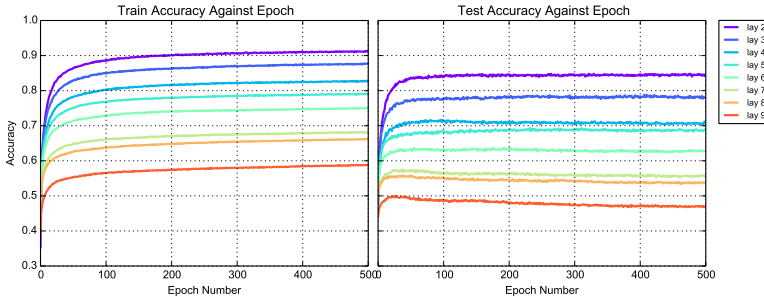

Figure 6: Demonstration of expressive power of remaining depth on MNIST. Here we plot train and test accuracy achieved by training exactly one layer of a fully connected neural net on MNIST. The different lines are generated by varying the hidden layer chosen to train. All other layers are kept frozen after random initialization. We see that training lower hidden layers leads to better performance. The networks had width $k = 100$, weight variance $\sigma_w^2 = 2$, and hard-tanh nonlinearities. Note that we only train from the second hidden layer (weights $W^{(1)}$) onwards, so that the number of parameters trained remains fixed. While the theory addresses training accuracy and not generalization accuracy, the same monotonic pattern is seen for both.

Given this class of functions, one useful measure of expressivity is determining how *heterogeneous* this class is. Inspired by classification tasks we formalize it as: given a set of inputs, $S = \{x_1, .., x_s\} \subset \mathbb{R}^m$, how many of the $2^s$ possible dichotomies does this function class produce on $S$?

For non-random inputs and non-random functions, this is a well known question upper bounded by the Sauer-Shelah lemma (Sauer, 1972). We discuss this further in Appendix D.1. In the random setting, the statistical duality of weight sweeping and input sweeping suggests a direct proportion to transitions and trajectory length for a fixed input. Furthermore, if the $x_i \in S$ are sufficiently uncorrelated (e.g. random) class label transitions should occur independently for each $x_i$ Indeed, we show this in Figure 4 (more figures, e.g. dichotomies vs transitions and observations, are included in the Appendix).

**Observation 2.** *Depth and Expressivity in a Function Class.* Given the function class $\mathcal{F}$ as above, the number of dichotomies expressible by $\mathcal{F}$ over a set of random inputs $S$ by sweeping the first layer weights along a one dimensional trajectory $W^{(0)}(t)$ is exponential in the network depth $n$.

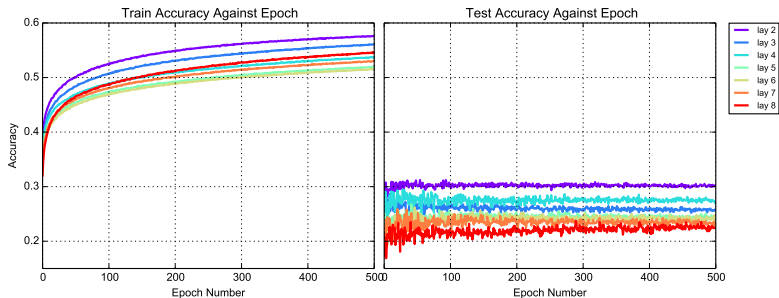

Figure 7: We repeat a similar experiment in Figure 6 with a fully connected network on CIFAR-10, and mostly observe that training lower layers again leads to better performance. The networks had width $k = 200$, weight variance $\sigma_w^2 = 1$, and hard-tanh nonlinearities. We again only train from the second hidden layer on so that the number of parameters remains fixed.

| Property | Architecture | Results |
|---|---|---|
| Trajectory length | hard-tanh | Asymptotically tight lower bound (Thm 1) Upper bound (Appendix Section A) Simulation (Fig 1) |
| Neuron transitions | hard-tanh | Expectation in large weight limit (Thm 2) Simulation (Fig 2) |
| Dichotomies | hard-tanh | Simulation (Figs 4 and 10) |
| Regions in input space | hard-tanh and ReLU | Consist of convex polytopes (Thm 3) |
| Network activation patterns | hard-tanh and ReLU | Tight upper bound (Thm 6) |
| Effect of remaining depth | hard-tanh | Simulation (Fig 5) Experiment on MNIST (Fig 6) Experiments on CIFAR-10 (Fig 7) |
| Effect of training on trajectory length | hard-tanh | Experiment on MNIST (Fig 8, 9) |

Table 1: List and location of key theoretical and experimental results.

## 3 TRAINED NETWORKS

**Remaining Depth**  The results from Section 2, particularly those linking dichotomies to trajectory length, suggest that earlier layers in the network might have more expressive power. In particular, the *remaining depth* of the network beyond the layer might directly influence its expressive power. We see that this holds in the random network case (Figure 5), and also for networks trained on MNIST and CIFAR-10. In Figures 6, 7 we randomly initialized a neural network, and froze all the layers except for one, which we trained.

**Training trades off between input-output map stability and expressivity**  We also look at the effect of training on measures of expressivity by plotting the change in trajectory length and number of transitions (see Appendix) during the training process. We find that for a network initialized with large $\sigma_w$, the training process appears to *stabilize* the input-output map – monotonically decreasing trajectory length (Figure 8) except for the final few steps. Interestingly, this happens at a faster rate in the vicinity of the data than for random inputs, and is accomplished *without* reducing weight magnitudes.

For a network closer to the boundary of the exponential regime $\sigma_w^2 = 3$, where trajectory length growth is still exponential but with a much smaller base, the training process *increases* the trajectory length, enabling greater expressivity in the resulting input-output map, Figure 9

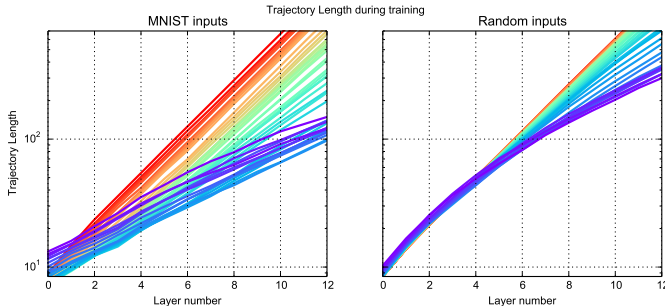

Figure 8: Training acts to stabilize the input-output map by decreasing trajectory length for $\sigma_w$ large. The left pane plots the growth of trajectory length as a circular interpolation between two MNIST datapoints is propagated through the network, at different train steps. Red indicates the start of training, with purple the end of training. Interestingly, and supporting the observation on remaining depth, the first layer appears to increase trajectory length, in contrast with all later layers, suggesting it is being primarily used to fit the data. The right pane shows an identical plot but for an interpolation between *random points*, which also display decreasing trajectory length, but at a slower rate. Note the output layer is not plotted, due to artificial scaling of length through normalization. The network is initialized with $\sigma_w^2 = 16$. A similar plot is observed for the number of transitions (see Appendix.)

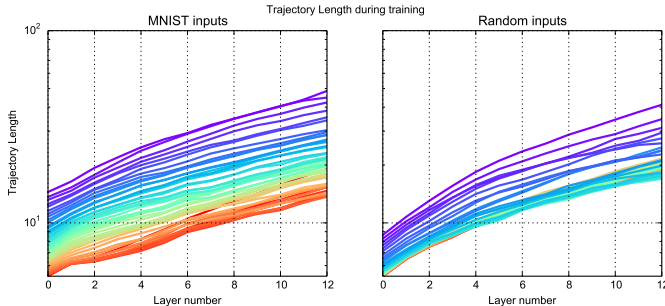

Figure 9: Training increases expressivity of input-output map for $\sigma_w$ small. The left pane plots the growth of trajectory length as a circular interpolation between two MNIST datapoints is propagated through the network, at different train steps. Red indicates the start of training, with purple the end of training. We see that the training process *increases* trajectory length, likely to increase the expressivity of the input-output map to enable greater accuracy. The right pane shows an identical plot but for an interpolation between *random points*, which also displays increasing trajectory length, but at a slower rate. Note the output layer is not plotted, due to artificial scaling of length through normalization. The network is initialized with $\sigma_w^2 = 3$.

## 4  CONCLUSION

In this paper, we studied the expressivity of neural networks through three measures, neuron transitions, activation patterns and dichotomies, and explained the observed exponential dependence on depth of all three measures by demonstrating the underlying link to latent trajectory length. Having explored these results in the context of random networks, we then looked at the consequences for trained networks (see Table 1). We find that the remaining depth above a network layer influences its expressive power, which might inspire new pre-training or initialization schemes. Furthermore, we see that training interpolates between expressive power and better generalization. This relation between initial and final parameters might inform early stopping and warm starting rules.

ACKNOWLEDGEMENTS

We thank Samy Bengio, Ian Goodfellow, Laurent Dinh, and Quoc Le for extremely helpful discussion.

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

# Appendix

Here we include the full proofs from sections in the paper.

## A  PROOFS AND ADDITIONAL RESULTS FROM SECTION 2.2

**Proof of Theorem 1**  We prove this result for $F_W$ with zero bias for technical simplicity. The result also translates over to $F_W$ with bias with a couple of technical modifications.

### A.1  NOTATION AND PRELIMINARY RESULTS

*Difference of points on trajectory* Given $x(t) = x, x(t + dt) = x + \delta x$ in the trajectory, let $\delta z^{(d)} = z^{(d)}(x + \delta x) - z^{(d)}(x)$

*Parallel and Perpendicular Components:* Given vectors $x, y$, we can write $y = y_\perp + y_\parallel$ where $y_\perp$ is the component of $y$ perpendicular to $x$, and $y_\parallel$ is the component parallel to $x$. (Strictly speaking, these components should also have a subscript $x$, but we suppress it as the direction with respect to which parallel and perpendicular components are being taken will be explicitly stated.)

This notation can also be used with a matrix $W$, see Lemma 1.

Before stating and proving the main theorem, we need a few preliminary results.

**Lemma 1.** Matrix Decomposition *Let $x, y \in \mathbb{R}^k$ be fixed non-zero vectors, and let $W$ be a (full rank) matrix. Then, we can write*

$$W = {}^\parallel W_\parallel + {}^\parallel W_\perp + {}^\perp W_\parallel + {}^\perp W_\perp$$

*such that*

$$
\begin{aligned}
{}^\parallel W_\perp x &= 0 & {}^\perp W_\perp x &= 0 \\
y^{T \perp} W_\parallel &= 0 & y^{T \perp} W_\perp &= 0
\end{aligned}
$$

*i.e. the row space of $W$ is decomposed to perpendicular and parallel components with respect to $x$ (subscript on right), and the column space is decomposed to perpendicular and parallel components of $y$ (superscript on left).*

*Proof.* Let $V, U$ be rotations such that $Vx = (\lVert x \rVert, 0..., 0)^T$ and $Uy = (\lVert y \rVert, 0...0)^T$. Now let $\tilde{W} = UWV^T$, and let $\tilde{W} = {}^\parallel \tilde{W}_\parallel + {}^\parallel \tilde{W}_\perp + {}^\perp \tilde{W}_\parallel + {}^\perp \tilde{W}_\perp$, with ${}^\parallel \tilde{W}_\parallel$ having non-zero term exactly $\tilde{W}_{11}$, ${}^\parallel \tilde{W}_\perp$ having non-zero entries exactly $\tilde{W}_{1i}$ for $2 \leq i \leq k$. Finally, we let ${}^\perp \tilde{W}_\parallel$ have non-zero entries exactly $\tilde{W}_{i1}$, with $2 \leq i \leq k$ and ${}^\perp \tilde{W}_\perp$ have the remaining entries non-zero.

If we define $\tilde{x} = Vx$ and $\tilde{y} = Uy$, then we see that

$$
\begin{aligned}
{}^\parallel \tilde{W}_\perp \tilde{x} &= 0 & {}^\perp \tilde{W}_\perp \tilde{x} &= 0 \\
\tilde{y}^{T \perp} \tilde{W}_\parallel &= 0 & \tilde{y}^{T \perp} \tilde{W}_\perp &= 0
\end{aligned}
$$

as $\tilde{x}, \tilde{y}$ have only one non-zero term, which does not correspond to a non-zero term in the components of $\tilde{W}$ in the equations.

Then, defining ${}^\parallel W_\parallel = U^{T \parallel} \tilde{W}_\parallel V$, and the other components analogously, we get equations of the form

$$ {}^\parallel W_\perp x = U^{T \parallel} \tilde{W}_\perp V x = U^{T \parallel} \tilde{W}_\perp \tilde{x} = 0 $$

$\square$

**Observation 3.** Given $W, x$ as before, and considering $W_\parallel, W_\perp$ with respect to $x$ (wlog a unit vector) we can express them directly in terms of $W$ as follows: Letting $W^{(i)}$ be the $i$th row of $W$, we have

$$W_\| = \begin{pmatrix} ((W^{(0)})^T \cdot x)x \\ \vdots \\ ((W^{(k)})^T \cdot x)x \end{pmatrix}$$

i.e. the projection of each row in the direction of $x$. And of course

$$W_\perp = W - W_\|$$

The motivation to consider such a decomposition of $W$ is for the resulting independence between different components, as shown in the following lemma.

**Lemma 2.** Independence of Projections *Let $x$ be a given vector (wlog of unit norm.) If $W$ is a random matrix with $W_{ij} \sim \mathcal{N}(0, \sigma^2)$, then $W_\|$ and $W_\perp$ with respect to $x$ are independent random variables.*

*Proof.* There are two possible proof methods:

(a) We use the rotational invariance of random Gaussian matrices, i.e. if $W$ is a Gaussian matrix, iid entries $\mathcal{N}(0, \sigma^2)$, and $R$ is a rotation, then $RW$ is also iid Gaussian, entries $\mathcal{N}(0, \sigma^2)$. (This follows easily from affine transformation rules for multivariate Gaussians.)

Let $V$ be a rotation as in Lemma 1. Then $\tilde{W} = WV^T$ is also iid Gaussian, and furthermore, $\tilde{W}_\|$ and $\tilde{W}_\perp$ partition the entries of $\tilde{W}$, so are evidently independent. But then $W_\| = \tilde{W}_\| V^T$ and $W_\perp = \tilde{W}_\perp V^T$ are also independent.

(b) From the observation note that $W_\|$ and $W_\perp$ have a centered multivariate joint Gaussian distribution (both consist of linear combinations of the entries $W_{ij}$ in $W$.) So it suffices to show that $W_\|$ and $W_\perp$ have covariance $0$. Because both are centered Gaussians, this is equivalent to showing $\mathbb{E}(< W_\|, W_\perp >) = 0$. We have that

$$\mathbb{E}(< W_\|, W_\perp >) = \mathbb{E}(W_\| W_\perp^T) = \mathbb{E}(W_\| W^T) - \mathbb{E}(W_\| W_\|^T)$$

As any two rows of $W$ are independent, we see from the observation that $\mathbb{E}(W_\| W^T)$ is a diagonal matrix, with the $i$th diagonal entry just $((W^{(0)})^T \cdot x)^2$. But similarly, $\mathbb{E}(W_\| W_\|^T)$ is also a diagonal matrix, with the same diagonal entries - so the claim follows.

$\square$

In the following two lemmas, we use the rotational invariance of Gaussians as well as the chi distribution to prove results about the expected norm of a random Gaussian vector.

**Lemma 3.** Norm of a Gaussian vector *Let $X \in \mathbb{R}^k$ be a random Gaussian vector, with $X_i$ iid, $\sim \mathcal{N}(0, \sigma^2)$. Then*

$$\mathbb{E}[||X||] = \sigma\sqrt{2}\frac{\Gamma((k+1)/2)}{\Gamma(k/2)}$$

*Proof.* We use the fact that if $Y$ is a random Gaussian, and $Y_i \sim \mathcal{N}(0, 1)$ then $||Y||$ follows a chi distribution. This means that $\mathbb{E}(||X/\sigma||) = \sqrt{2}\Gamma((k+1)/2)/\Gamma(k/2)$, the mean of a chi distribution with $k$ degrees of freedom, and the result follows by noting that the expectation in the lemma is $\sigma$ multiplied by the above expectation. $\square$

We will find it useful to bound ratios of the Gamma function (as appear in Lemma 3) and so introduce the following inequality, from (Kershaw, 1983) that provides an extension of Gautschi's Inequality.

**Theorem 5.** An Extension of Gautschi's Inequality *For $0 < s < 1$, we have*

$$\left(x + \frac{s}{2}\right)^{1-s} \leq \frac{\Gamma(x+1)}{\Gamma(x+s)} \leq \left(x - \frac{1}{2} + \left(s + \frac{1}{4}\right)^{\frac{1}{2}}\right)^{1-s}$$

We now show:

**Lemma 4.** Norm of Projections *Let $W$ be a $k$ by $k$ random Gaussian matrix with iid entries $\sim \mathcal{N}(0, \sigma^2)$, and $x, y$ two given vectors. Partition $W$ into components as in Lemma 1 and let $x_\perp$ be a nonzero vector perpendicular to $x$. Then*

*(a)*

$$\mathbb{E}\left[||^\perp W_\perp x_\perp||\right] = ||x_\perp|| \, \sigma\sqrt{2} \frac{\Gamma(k/2)}{\Gamma((k-1)/2)} \geq ||x_\perp|| \, \sigma\sqrt{2} \left(\frac{k}{2} - \frac{3}{4}\right)^{1/2}$$

*(b) If $\mathbb{1}_\mathcal{A}$ is an identity matrix with non-zeros diagonal entry $i$ iff $i \in \mathcal{A} \subset [k]$, and $|A| > 2$, then*

$$\mathbb{E}\left[||\mathbb{1}_\mathcal{A}{}^\perp W_\perp x_\perp||\right] \geq ||x_\perp|| \, \sigma\sqrt{2} \frac{\Gamma(|\mathcal{A}|/2)}{\Gamma((|\mathcal{A}|-1)/2)} \geq ||x_\perp|| \, \sigma\sqrt{2} \left(\frac{|\mathcal{A}|}{2} - \frac{3}{4}\right)^{1/2}$$

*Proof.* (a) Let $U, V, \tilde{W}$ be as in Lemma 1. As $U, V$ are rotations, $\tilde{W}$ is also iid Gaussian. Furthermore for any fixed $W$, with $\tilde{a} = Va$, by taking inner products, and square-rooting, we see that $\left||\tilde{W}\tilde{a}\right|| = ||Wa||$. So in particular

$$\mathbb{E}\left[||^\perp W_\perp x_\perp||\right] = \mathbb{E}\left[\left||^\perp \tilde{W}_\perp \tilde{x}_\perp\right||\right]$$

But from the definition of non-zero entries of $^\perp\tilde{W}_\perp$, and the form of $\tilde{x}_\perp$ (a zero entry in the first coordinate), it follows that $^\perp\tilde{W}_\perp\tilde{x}_\perp$ has exactly $k-1$ non zero entries, each a centered Gaussian with variance $(k-1)\sigma^2 ||x_\perp||^2$. By Lemma 3, the expected norm is as in the statement. We then apply Theorem 5 to get the lower bound.

(b) First note we can view $\mathbb{1}_\mathcal{A}{}^\perp W_\perp = {}^\perp\mathbb{1}_\mathcal{A}W_\perp$. (Projecting down to a random (as $W$ is random) subspace of fixed size $|\mathcal{A}| = m$ and then making perpendicular commutes with making perpendicular and then projecting everything down to the subspace.)

So we can view $W$ as a random $m$ by $k$ matrix, and for $x, y$ as in Lemma 1 (with $y$ projected down onto $m$ dimensions), we can again define $U, V$ as $k$ by $k$ and $m$ by $m$ rotation matrices respectively, and $\tilde{W} = UWV^T$, with analogous properties to Lemma 1. Now we can finish as in part (a), except that $^\perp\tilde{W}_\perp\tilde{x}$ may have only $m-1$ entries, (depending on whether $y$ is annihilated by projecting down by $\mathbb{1}_\mathcal{A}$) each of variance $(k-1)\sigma^2 ||x_\perp||^2$.

$\square$

**Lemma 5.** Norm and Translation *Let $X$ be a centered multivariate Gaussian, with diagonal covariance matrix, and $\mu$ a constant vector.*

$$\mathbb{E}(||X - \mu||) \geq \mathbb{E}(||X||)$$

*Proof.* The inequality can be seen intuitively geometrically: as $X$ has diagonal covariance matrix, the contours of the pdf of $||X||$ are circular centered at 0, decreasing radially. However, the contours of the pdf of $||X - \mu||$ are shifted to be centered around $||\mu||$, and so shifting back $\mu$ to 0 reduces the norm.

A more formal proof can be seen as follows: let the pdf of $X$ be $f_X(\cdot)$. Then we wish to show

$$\int_x ||x - \mu|| \, f_X(x)dx \geq \int_x ||x|| \, f_X(x)dx$$

Now we can pair points $x, -x$, using the fact that $f_X(x) = f_X(-x)$ and the triangle inequality on the integrand to get

$$\int_{|x|} (||x - \mu|| + ||-x - \mu||) \, f_X(x)dx \geq \int_{|x|} ||2x|| \, f_X(x)dx = \int_{|x|} (||x|| + ||-x||) \, f_X(x)dx$$

$\square$

A.2 PROOF OF THEOREM

*Proof.* We first prove the zero bias case, Theorem 1. To do so, it is sufficient to prove that

$$\mathbb{E}\left[\left|\left|\delta z^{(d+1)}(t)\right|\right|\right] \geq O\left(\left(\frac{\sqrt{\sigma k}}{\sqrt{\sigma + k}}\right)^{d+1}\right)\left|\left|\delta z^{(0)}(t)\right|\right| \tag{**}$$

as integrating over $t$ gives us the statement of the theorem.

For ease of notation, we will suppress the $t$ in $z^{(d)}(t)$.

We first write

$$W^{(d)} = W_{\perp}^{(d)} + W_{\parallel}^{(d)}$$

where the division is done with respect to $z^{(d)}$. Note that this means $h^{(d+1)} = W_{\parallel}^{(d)} z^{(d)}$ as the other component annihilates (maps to 0) $z^{(d)}$.

We can also define $\mathcal{A}_{W_{\parallel}^{(d)}} = \{i : i \in [k], |h_i^{(d+1)}| < 1\}$ i.e. the set of indices for which the hidden representation is not saturated. Letting $W_i$ denote the $i$th row of matrix $W$, we now claim that:

$$\mathbb{E}_{W^{(d)}}\left[\left|\left|\delta z^{(d+1)}\right|\right|\right] = \mathbb{E}_{W_{\parallel}^{(d)}}\mathbb{E}_{W_{\perp}^{(d)}}\left[\left(\sum_{i \in \mathcal{A}_{W_{\parallel}^{(d)}}}((W_{\perp}^{(d)})_i \delta z^{(d)} + (W_{\parallel}^{(d)})_i \delta z^{(d)})^2\right)^{1/2}\right] \tag{*}$$

Indeed, by Lemma 2 we first split the expectation over $W^{(d)}$ into a tower of expectations over the two independent parts of $W$ to get

$$\mathbb{E}_{W^{(d)}}\left[\left|\left|\delta z^{(d+1)}\right|\right|\right] = \mathbb{E}_{W_{\parallel}^{(d)}}\mathbb{E}_{W_{\perp}^{(d)}}\left[\left|\left|\phi(W^{(d)}\delta z^{(d)})\right|\right|\right]$$

But conditioning on $W_{\parallel}^{(d)}$ in the inner expectation gives us $h^{(d+1)}$ and $\mathcal{A}_{W_{\parallel}^{(d)}}$, allowing us to replace the norm over $\phi(W^{(d)}\delta z^{(d)})$ with the sum in the term on the right hand side of the claim.

Till now, we have mostly focused on partitioning the matrix $W^{(d)}$. But we can also set $\delta z^{(d)} = \delta z_{\parallel}^{(d)} + \delta z_{\perp}^{(d)}$ where the perpendicular and parallel are with respect to $z^{(d)}$. In fact, to get the expression in (**), we derive a recurrence as below:

$$\mathbb{E}_{W^{(d)}}\left[\left|\left|\delta z_{\perp}^{(d+1)}\right|\right|\right] \geq O\left(\frac{\sqrt{\sigma k}}{\sqrt{\sigma + k}}\right)\mathbb{E}_{W^{(d)}}\left[\left|\left|\delta z_{\perp}^{(d)}\right|\right|\right]$$

To get this, we first need to define $\tilde{z}^{(d+1)} = \mathbb{1}_{\mathcal{A}_{W_{\parallel}^{(d)}}} h^{(d+1)}$ - the latent vector $h^{(d+1)}$ with all saturated units zeroed out.

We then split the column space of $W^{(d)} = {}^{\perp}W^{(d)} + {}^{\parallel}W^{(d)}$, where the split is with respect to $\tilde{z}^{(d+1)}$. Letting $\delta z_{\perp}^{(d+1)}$ be the part perpendicular to $z^{(d+1)}$, and $\mathcal{A}$ the set of units that are unsaturated, we have an important relation:

**Claim**

$$\left|\left|\delta z_{\perp}^{(d+1)}\right|\right| \geq \left|\left|{}^{\perp}W^{(d)}\delta z^{(d)}\mathbb{1}_{\mathcal{A}}\right|\right|$$

(where the indicator in the right hand side zeros out coordinates not in the active set.)

To see this, first note, by definition,

$$\delta z_{\perp}^{(d+1)} = W^{(d)}\delta z^{(d)} \cdot \mathbb{1}_{\mathcal{A}} - \langle W^{(d)}\delta z^{(d)} \cdot \mathbb{1}_{\mathcal{A}}, \hat{z}^{(d+1)}\rangle\hat{z}^{(d+1)} \tag{1}$$

where the $\hat{\cdot}$ indicates a unit vector.

Similarly

$$^\perp W^{(d)} \delta z^{(d)} = W^{(d)} \delta z^{(d)} - \langle W^{(d)} \delta z^{(d)}, \hat{z}^{(d+1)} \rangle \hat{z}^{(d+1)} \tag{2}$$

Now note that for any index $i \in \mathcal{A}$, the right hand sides of (1) and (2) are identical, and so the vectors on the left hand side agree for all $i \in \mathcal{A}$. In particular,

$$\delta z_\perp^{(d+1)} \cdot \mathbb{1}_\mathcal{A} = {}^\perp W^{(d)} \delta z^{(d)} \cdot \mathbb{1}_\mathcal{A}$$

Now the claim follows easily by noting that $\left\| \delta z_\perp^{(d+1)} \right\| \geq \left\| \delta z_\perp^{(d+1)} \cdot \mathbb{1}_\mathcal{A} \right\|$.

Returning to (*), we split $\delta z^{(d)} = \delta z_\perp^{(d)} + \delta z_\parallel^{(d)}$, $W_\perp^{(d)} = {}^\parallel W_\perp^{(d)} + {}^\perp W_\perp^{(d)}$ (and $W_\parallel^{(d)}$ analogously), and after some cancellation, we have

$$\mathbb{E}_{W^{(d)}} \left[ \left\| \delta z^{(d+1)} \right\| \right] = \mathbb{E}_{W_\parallel^{(d)}} \mathbb{E}_{W_\perp^{(d)}} \left[ \left( \sum_{i \in \mathcal{A}_{W_\parallel^{(d)}}} \left( ({}^\perp W_\perp^{(d)} + {}^\parallel W_\perp^{(d)})_i \delta z_\perp^{(d)} + ({}^\perp W_\parallel^{(d)} + {}^\parallel W_\parallel^{(d)})_i \delta z_\parallel^{(d)} \right)^2 \right)^{1/2} \right]$$

We would like a recurrence in terms of only perpendicular components however, so we first drop the ${}^\parallel W_\perp^{(d)}, {}^\parallel W_\parallel^{(d)}$ (which can be done without decreasing the norm as they are perpendicular to the remaining terms) and using the above claim, have

$$\mathbb{E}_{W^{(d)}} \left[ \left\| \delta z_\perp^{(d+1)} \right\| \right] \geq \mathbb{E}_{W_\parallel^{(d)}} \mathbb{E}_{W_\perp^{(d)}} \left[ \left( \sum_{i \in \mathcal{A}_{W_\parallel^{(d)}}} \left( ({}^\perp W_\perp^{(d)})_i \delta z_\perp^{(d)} + ({}^\perp W_\parallel^{(d)})_i \delta z_\parallel^{(d)} \right)^2 \right)^{1/2} \right]$$

But in the inner expectation, the term ${}^\perp W_\parallel^{(d)} \delta z_\parallel^{(d)}$ is just a constant, as we are conditioning on $W_\parallel^{(d)}$. So using Lemma 5 we have

$$\mathbb{E}_{W_\perp^{(d)}} \left[ \left( \sum_{i \in \mathcal{A}_{W_\parallel^{(d)}}} \left( ({}^\perp W_\perp^{(d)})_i \delta z_\perp^{(d)} + ({}^\perp W_\parallel^{(d)})_i \delta z_\parallel^{(d)} \right)^2 \right)^{1/2} \right] \geq \mathbb{E}_{W_\perp^{(d)}} \left[ \left( \sum_{i \in \mathcal{A}_{W_\parallel^{(d)}}} \left( ({}^\perp W_\perp^{(d)})_i \delta z_\perp^{(d)} \right)^2 \right)^{1/2} \right]$$

We can then apply Lemma 4 to get

$$\mathbb{E}_{W_\perp^{(d)}} \left[ \left( \sum_{i \in \mathcal{A}_{W_\parallel^{(d)}}} \left( ({}^\perp W_\perp^{(d)})_i \delta z_\perp^{(d)} \right)^2 \right)^{1/2} \right] \geq \frac{\sigma}{\sqrt{k}} \sqrt{2} \frac{\sqrt{2|\mathcal{A}_{W_\parallel^{(d)}}| - 3}}{2} \mathbb{E} \left[ \left\| \delta z_\perp^{(d)} \right\| \right]$$

The outer expectation on the right hand side only affects the term in the expectation through the size of the non-saturated set of units. Letting $p = \mathbb{P}(|h_i^{(d+1)}| < 1)$, and noting that we get a non-zero norm only if $|\mathcal{A}_{W_\parallel^{(d)}}| \geq 2$ (else we cannot project down a dimension), and for $|\mathcal{A}_{W_\parallel^{(d)}}| \geq 2$,

$$\sqrt{2} \frac{\sqrt{2|\mathcal{A}_{W_\parallel^{(d)}}| - 3}}{2} \geq \frac{1}{\sqrt{2}} \sqrt{|\mathcal{A}_{W_\parallel^{(d)}}|}$$

we get

$$\mathbb{E}_{W^{(d)}} \left[ \left\| \delta z_\perp^{(d+1)} \right\| \right] \geq \frac{1}{\sqrt{2}} \left( \sum_{j=2}^{k} \binom{k}{j} p^j (1-p)^{k-j} \frac{\sigma}{\sqrt{k}} \sqrt{j} \right) \mathbb{E} \left[ \left\| \delta z_\perp^{(d)} \right\| \right]$$

We use the fact that we have the probability mass function for an $(k, p)$ binomial random variable to bound the $\sqrt{j}$ term:

$$\sum_{j=2}^{k} \binom{k}{j} p^j (1-p)^{k-j} \frac{\sigma}{\sqrt{k}} \sqrt{j} = -\binom{k}{1} p(1-p)^{k-1} \frac{\sigma}{\sqrt{k}} + \sum_{j=0}^{k} \binom{k}{j} p^j (1-p)^{k-j} \frac{\sigma}{\sqrt{k}} \sqrt{j}$$

$$= -\sigma \sqrt{k} p (1-p)^{k-1} + kp \cdot \frac{\sigma}{\sqrt{k}} \sum_{j=1}^{k} \frac{1}{\sqrt{j}} \binom{k-1}{j-1} p^{j-1} (1-p)^{k-j}$$

But by using Jensen's inequality with $1/\sqrt{x}$, we get

$$\sum_{j=1}^{k} \frac{1}{\sqrt{j}} \binom{k-1}{j-1} p^{j-1} (1-p)^{k-j} \geq \frac{1}{\sqrt{\sum_{j=1}^{k} j \binom{k-1}{j-1} p^{j-1} (1-p)^{k-j}}} = \frac{1}{\sqrt{(k-1)p+1}}$$

where the last equality follows by recognising the expectation of a binomial$(k-1, p)$ random variable. So putting together, we get

$$\mathbb{E}_{W^{(d)}} \left[ \left\| \delta z_\perp^{(d+1)} \right\| \right] \geq \frac{1}{\sqrt{2}} \left( -\sigma \sqrt{k} p (1-p)^{k-1} + \sigma \cdot \frac{\sqrt{k}p}{\sqrt{1+(k-1)p}} \right) \mathbb{E} \left[ \left\| \delta z_\perp^{(d)} \right\| \right] \quad \text{(a)}$$

To lower bound $p$, we first note that as $h_i^{(d+1)}$ is a normal random variable with variance $\leq \sigma^2$, if $A \sim \mathcal{N}(0, \sigma^2)$

$$\mathbb{P}(|h_i^{(d+1)}| < 1) \geq \mathbb{P}(|A| < 1) \geq \frac{1}{\sigma \sqrt{2\pi}} \quad \text{(b)}$$

where the last inequality holds for $\sigma \geq 1$ and follows by Taylor expanding $e^{-x^2/2}$ around 0. Similarly, we can also show that $p \leq \frac{1}{\sigma}$.

So this becomes

$$\mathbb{E} \left[ \left\| \delta z^{(d+1)} \right\| \right] \geq \left( \frac{1}{\sqrt{2}} \left( \frac{1}{(2\pi)^{1/4}} \frac{\sqrt{\sigma k}}{\sqrt{\sigma \sqrt{2\pi} + (k-1)}} - \sqrt{k} \left( 1 - \frac{1}{\sigma} \right)^{k-1} \right) \right) \mathbb{E} \left[ \left\| \delta z_\perp^{(d)} \right\| \right]$$

$$= O \left( \frac{\sqrt{\sigma k}}{\sqrt{\sigma + k}} \right) \mathbb{E} \left[ \left\| \delta z_\perp^{(d)} \right\| \right]$$

Finally, we can compose this, to get

$$\mathbb{E} \left[ \left\| \delta z^{(d+1)} \right\| \right] \geq \left( \frac{1}{\sqrt{2}} \left( \frac{1}{(2\pi)^{1/4}} \frac{\sqrt{\sigma k}}{\sqrt{\sigma \sqrt{2\pi} + (k-1)}} - \sqrt{k} \left( 1 - \frac{1}{\sigma} \right)^{k-1} \right) \right)^{d+1} c \cdot \|\delta x(t)\|$$

with the constant $c$ being the ratio of $\|\delta x(t)_\perp\|$ to $\|\delta x(t)\|$. So if our trajectory direction is almost orthogonal to $x(t)$ (which will be the case for e.g. random circular arcs, $c$ can be seen to be $\approx 1$ by splitting into components as in Lemma 1, and using Lemmas 3, 4.)

$\square$

**Result for non-zero bias** In fact, we can easily extend the above result to the case of non-zero bias. The insight is to note that because $\delta z^{(d+1)}$ involves taking a *difference* between $z^{(d+1)}(t+dt)$ and $z^{(d+1)}(t)$, the bias term does not enter at all into the expression for $\delta z^{(d+1)}$. So the computations above hold, and equation (a) becomes

$$\mathbb{E}_{W^{(d)}} \left[ \left\| \delta z_\perp^{(d+1)} \right\| \right] \geq \frac{1}{\sqrt{2}} \left( -\sigma_w \sqrt{k} p (1-p)^{k-1} + \sigma_w \cdot \frac{\sqrt{k}p}{\sqrt{1+(k-1)p}} \right) \mathbb{E} \left[ \left\| \delta z_\perp^{(d)} \right\| \right]$$

We also now have that $h_i^{(d+1)}$ is a normal random variable with variance $\leq \sigma_w^2 + \sigma_b^2$ (as the bias is drawn from $\mathcal{N}(0, \sigma_b^2)$). So equation (b) becomes

$$\mathbb{P}(|h_i^{(d+1)}| < 1) \geq \frac{1}{\sqrt{(\sigma_w^2 + \sigma_b^2)}\sqrt{2\pi}}$$

This gives Theorem 1

$$\mathbb{E}\left[\left\|\left|\delta z^{(d+1)}\right|\right\|\right] \geq O\left(\frac{\sigma_w}{(\sigma_w^2 + \sigma_b^2)^{1/4}} \cdot \frac{\sqrt{k}}{\sqrt{\sqrt{\sigma_w^2 + \sigma_b^2} + k}}\right) \mathbb{E}\left[\left\|\left|\delta z_\perp^{(d)}\right|\right\|\right]$$

**Statement and Proof of Upper Bound for Trajectory Growth**   Replace hard-tanh with a linear coordinate-wise identity map, $h_i^{(d+1)} = (W^{(d)} z^{(d)})_i + b_i$. This provides an upper bound on the norm. We also then recover a chi distribution with $k$ terms, each with standard deviation $\frac{\sigma_w}{k^{\frac{1}{2}}}$,

$$\mathbb{E}\left[\left\|\left|\delta z^{(d+1)}\right|\right\|\right] \leq \sqrt{2}\frac{\Gamma\left((k+1)/2\right)}{\Gamma\left(k/2\right)}\frac{\sigma_w}{k^{\frac{1}{2}}}\left\|\left|\delta z^{(d)}\right|\right\| \tag{2}$$

$$\leq \sigma_w\left(\frac{k+1}{k}\right)^{\frac{1}{2}}\left\|\left|\delta z^{(d)}\right|\right\|, \tag{3}$$

where the second step follows from (Laforgia and Natalini, 2013), and holds for $k > 1$.

# B   PROOFS AND ADDITIONAL RESULTS FROM SECTION 2.2.2

**Proof of Theorem 2**

*Proof.* **For $\sigma_b = 0$:**

For hidden layer $d < n$, consider neuron $v_1^{(d)}$. This has as input $\sum_{i=1}^k W_{i1}^{(d-1)} z_i^{(d-1)}$. As we are in the large $\sigma$ case, we assume that $|z_i^{(d-1)}| = 1$. Furthermore, as signs for $z_i^{(d-1)}$ and $W_{i1}^{(d-1)}$ are both completely random, we can also assume wlog that $z_i^{(d-1)} = 1$. For a particular input, we can define $v_1^{(d)}$ as *sensitive* to $v_i^{(d-1)}$ if $v_i^{(d-1)}$ transitioning (to wlog $-1$) will induce a transition in node $v_1^{(d)}$. A sufficient condition for this to happen is if $|W_{i1}| \geq |\sum_{j\neq i} W_{j1}|$. But $X = W_{i1} \sim \mathcal{N}(0, \sigma^2/k)$ and $\sum_{j\neq i} W_{j1} = Y' \sim \mathcal{N}(0, (k-1)\sigma^2/k)$. So we want to compute $\mathbb{P}(|X| > |Y'|)$. For ease of computation, we instead look at $\mathbb{P}(|X| > |Y|)$, where $Y \sim \mathcal{N}(0, \sigma^2)$.

But this is the same as computing $\mathbb{P}(|X|/|Y| > 1) = \mathbb{P}(X/Y < -1) + \mathbb{P}(X/Y > 1)$. But the ratio of two centered independent normals with variances $\sigma_1^2, \sigma_2^2$ follows a Cauchy distribution, with parameter $\sigma_1/\sigma_2$, which in this case is $1/\sqrt{k}$. Substituting this in to the cdf of the Cauchy distribution, we get that

$$\mathbb{P}\left(\frac{|X|}{|Y|} > 1\right) = 1 - \frac{2}{\pi}\arctan(\sqrt{k})$$

Finally, using the identity $\arctan(x) + \arctan(1/x)$ and the Laurent series for $\arctan(1/x)$, we can evaluate the right hand side to be $O(1/\sqrt{k})$. In particular

$$\mathbb{P}\left(\frac{|X|}{|Y|} > 1\right) \geq O\left(\frac{1}{\sqrt{k}}\right) \tag{c}$$

This means that in expectation, any neuron in layer $d$ will be sensitive to the transitions of $\sqrt{k}$ neurons in the layer below. Using this, and the fact the while $v_i^{(d-1)}$ might flip very quickly from say $-1$ to $1$, the gradation in the transition ensures that neurons in layer $d$ sensitive to $v_i^{(d-1)}$ will transition at distinct times, we get the desired growth rate in expectation as follows:

Let $T^{(d)}$ be a random variable denoting the number of transitions in layer $d$. And let $T_i^{(d)}$ be a random variable denoting the number of transitions of neuron $i$ in layer $d$. Note that by linearity of expectation and symmetry, $\mathbb{E}\left[T^{(d)}\right] = \sum_i \mathbb{E}\left[T_i^{(d)}\right] = k\mathbb{E}\left[T_1^{(d)}\right]$

Now, $\mathbb{E}\left[T_1^{(d+1)}\right] \geq \mathbb{E}\left[\sum_i 1_{(1,i)} T_i^{(d)}\right] = k\mathbb{E}\left[1_{(1,1)} T_1^{(d)}\right]$ where $1_{(1,i)}$ is the indicator function of neuron 1 in layer $d+1$ being sensitive to neuron $i$ in layer $d$.

But by the independence of these two events, $\mathbb{E}\left[1_{(1,1)} T_1^{(d)}\right] = \mathbb{E}\left[1_{(1,1)}\right] \cdot \mathbb{E}\left[T_1^{(d)}\right]$. But the firt time on the right hand side is $O(1/\sqrt{k})$ by (c), so putting it all together, $\mathbb{E}\left[T_1^{(d+1)}\right] \geq \sqrt{k}\mathbb{E}\left[T_1^{(d)}\right]$.

Written in terms of the entire layer, we have $\mathbb{E}\left[T^{(d+1)}\right] \geq \sqrt{k}\mathbb{E}\left[T^{(d)}\right]$ as desired.

**For $\sigma_b > 0$:**

We replace $\sqrt{k}$ with $\sqrt{k(1 + \sigma_b^2/\sigma_w^2)}$, by noting that $Y \sim \mathcal{N}(0, \sigma_w^2 + \sigma_b^2)$. This results in a growth rate of form $O(\sqrt{k}/\sqrt{1 + \frac{\sigma_b^2}{\sigma_w^2}})$. $\qquad\square$

## C    PROOFS AND ADDITIONAL RESULTS FROM SECTION 2.3

**Proof of Theorem 3**

*Proof.* We show inductively that $F_W$ partitions the input space into convex polytopes via hyperplanes. Consider the image of the input space under the first hidden layer. Each neuron $v_i^{(1)}$ defines hyperplane(s) on the input space: letting $W_i^{(0)}$ be the $i$th row of $W^{(0)}$, $b_i^{(0)}$ the bias, we have the hyperplane $W_i^{(0)}x + b_i = 0$ for a ReLU and hyperplanes $W_i^{(0)}x + b_i = \pm 1$ for a hard-tanh. Considering all such hyperplanes over neurons in the first layer, we get a hyperplane arrangement in the input space, each polytope corresponding to a specific activation pattern in the first hidden layer.

Now, assume we have partitioned our input space into convex polytopes with hyperplanes from layers $\leq d-1$. Consider $v_i^{(d)}$ and a specific polytope $R_i$. Then the activation pattern on layers $\leq d-1$ is constant on $R_i$, and so the input to $v_i^{(d)}$ on $R_i$ is a linear function of the inputs $\sum_j \lambda_j x_j + b$ and some constant term, comprising of the bias and the output of saturated units. Setting this expression to zero (for ReLUs) or to $\pm 1$ (for hard-tanh) again gives a hyperplane equation, but this time, the equation is only valid in $R_i$ (as we get a different linear function of the inputs in a different region.) So the defined hyperplane(s) either partition $R_i$ (if they intersect $R_i$) or the output pattern of $v_i^{(d)}$ is also constant on $R_i$. The theorem then follows. $\qquad\square$

This implies that any one dimensional trajectory $x(t)$, that does not 'double back' on itself (i.e. reenter a polytope it has previously passed through), will not repeat activation patterns. In particular, after seeing a transition (crossing a hyperplane to a different region in input space) we will never return to the region we left. A simple example of such a trajectory is a straight line:

**Corollary 2.** Transitions and Output Patterns in an Affine Trajectory *For any affine one dimensional trajectory $x(t) = x_0 + t(x_1 - x_0)$ input into a neural network $F_W$, we partition $\mathbb{R} \ni t$ into intervals every time a neuron transitions. Every interval has a unique network activation pattern on $F_W$.*

Generalizing from a one dimensional trajectory, we can ask how many regions are achieved over the entire input – i.e. how many distinct activation patterns are seen? We first prove a bound on the number of regions formed by $k$ hyperplanes in $\mathbb{R}^m$ (in a purely elementary fashion, unlike the proof presented in (Stanley, 2011))

**Theorem 6.** Upper Bound on Regions in a Hyperplane Arrangement *Suppose we have $k$ hyperplanes in $\mathbb{R}^m$ - i.e. $k$ equations of form $\alpha_i x = \beta_i$. for $\alpha_i \in \mathbb{R}^m$, $\beta_i \in \mathbb{R}$. Let the number of regions (connected open sets bounded on some sides by the hyperplanes) be $r(k,m)$. Then*

$$r(k,m) \leq \sum_{i=0}^{m} \binom{k}{i}$$

**Proof of Theorem 6**

*Proof.* Let the hyperplane arrangement be denoted $\mathcal{H}$, and let $H \in \mathcal{H}$ be one specific hyperplane. Then the number of regions in $\mathcal{H}$ is precisely the number of regions in $\mathcal{H} - H$ plus the number of

regions in $\mathcal{H} \cap H$. (This follows from the fact that $H$ subdivides into two regions exactly all of the regions in $\mathcal{H} \cap H$, and does not affect any of the other regions.)

In particular, we have the recursive formula

$$r(k, m) = r(k - 1, m) + r(k - 1, m - 1)$$

We now induct on $k + m$ to assert the claim. The base cases of $r(1, 0) = r(0, 1) = 1$ are trivial, and assuming the claim for $\leq k + m - 1$ as the induction hypothesis, we have

$$r(k - 1, m) + r(k - 1, m - 1) \leq \sum_{i=0}^{m} \binom{k-1}{i} + \sum_{i=0}^{m-1} \binom{k-1}{i}$$

$$\leq \binom{k-1}{0} + \sum_{i=0}^{d-1} \binom{k-1}{i} + \binom{k-1}{i+1}$$

$$\leq \binom{k}{0} + \sum_{i=0}^{m-1} \binom{k}{i+1}$$

where the last equality follows by the well known identity

$$\binom{a}{b} + \binom{a}{b+1} = \binom{a+1}{b+1}$$

This concludes the proof. $\qquad \square$

With this result, we can easily prove Theorem 4 as follows:

*Proof.* First consider the ReLU case. Each neuron has one hyperplane associated with it, and so by Theorem 6, the first hidden layer divides up the inputs space into $r(k, m)$ regions, with $r(k, m) \leq O(k^m)$.

Now consider the second hidden layer. For every region in the first hidden layer, there is a different activation pattern in the first layer, and so (as described in the proof of Theorem 3) a different hyperplane arrangement of $k$ hyperplanes in an $m$ dimensional space, contributing at most $r(k, m)$ regions.

In particular, the total number of regions in input space as a result of the first and second hidden layers is $\leq r(k, m) * r(k, m) \leq O(k^2 m)$. Continuing in this way for each of the $n$ hidden layers gives the $O(k^m n)$ bound.

A very similar method works for hard tanh, but here each neuron produces two hyperplanes, resulting in a bound of $O((2k)^{mn})$.

$\qquad \square$

# D    PROOFS AND ADDITIONAL RESULTS FROM SECTION 2.4

## D.1    UPPER BOUND FOR DICHOTOMIES

The Vapnik-Chervonenkis (VC) dimension of a function class is the cardinality of the largest set of points that it can shatter. The VC dimension provides an upper (worst case) bound on the generalization error for a function class (Vapnik and Vapnik, 1998). Motivated by generalization error, VC dimension has been studied for neural networks (Sontag, 1998; Bartlett and Maass, 2003). In (Bartlett et al., 1998) an upper bound on the VC dimension $v$ of a neural network with piecewise polynomial activation function and binary output is derived. For hard-tanh units, this bound is

$$v = 2 |W| n \log (4e |W| nk) + 2 |W| n^2 \log 2 + 2n, \tag{4}$$

where $|W|$ is the total number of weights, $n$ is the depth, and $k$ is the width of the network. The VC dimension provides an upper bound on the number of achievable dichotomies $|\mathcal{F}|$ by way of the

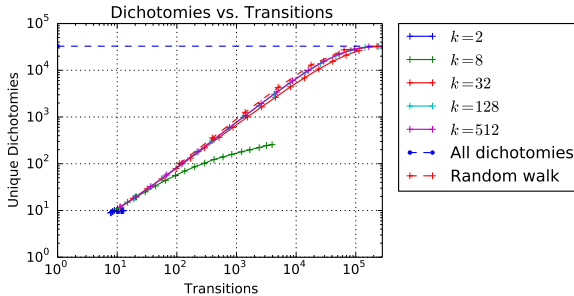

Figure 10: Here we plot the number of unique dichotomies that have been observed as a function of the number of transitions the network has undergone. Each datapoint corresponds to the number of transitions and dichotomies for a hard-tanh network of a different depth, with the weights in the first layer undergoing interpolation along a great circle trajectory $W^{(0)}(t)$. We compare these plots to a random walk simulation, where at each transition a single class label is flipped uniformly at random. Dichotomies are measured over a dataset consisting of $s = 15$ random samples, and all networks had weight variance $\sigma_w^2 = 16$. The blue dashed line indicates all $2^s$ possible dichotomies.

Sauer–Shelah lemma (Sauer, 1972),

$$|\mathcal{F}| \leq \left(\frac{e|S|}{v}\right)^v. \tag{5}$$

By combining Equations 4 and 5 an upper bound on the number of dichotomies is found, with a growth rate which is exponential in a low order polynomial of the network size.

Our results further suggest the following conjectures:

**Conjecture 1.** *As network width $k$ increases, the exploration of the space of dichotomies increasingly resembles a simple random walk on a hypercube with dimension equal to the number of inputs $|S|$.*

This conjecture is supported by Figure 10, which compares the number of unique dichotomies achieved by networks of various widths to the number of unique dichotomies achieved by a random walk. This is further supported by an exponential decrease in autocorrelation length in function space, derived in our prior work (Poole et al., 2016).

**Conjecture 2.** *The expressive power of a single weight $W_{ij}^{(d)}$ at layer $d$ in a random network $F$, and for a set of random inputs $S$, is exponential in the remaining network depth $d_r = (n - d)$. Here expressive power is the number of dichotomies achievable by adjusting only that weight.*

That is, the expressive power of weights in early layers in a deep hard-tanh network is exponentially greater than the expressive power of weights in later layers. This is supported by the invariance to layer number in the recurrence relations used in all proofs directly involving depth. It is also directly supported by simulation, as illustrated in Figure 5, and by experiments on MNIST and CIFAR10 as illustrated in Figures 6, 7.

## E FURTHER RESULTS AND IMPLEMENTATION DETAILS FROM SECTION 3

We implemented the random network architecture described in Section 2.1. In separate experiments we then swept an input vector along a great circle trajectory (a rotation) for fixed weights, and swept weights along a great circle trajectory for a fixed set of inputs, as described in Section 2.4. In both cases, the trajectory was subdivided into $10^6$ segments. We repeated this for a grid of network widths $k$, weight variances $\sigma_w^2$, and number of inputs $s$. Unless otherwise noted, $\sigma_b = 0$ for all experiments. We repeated each experiment 10 times and averaged over the results. The simulation results are discussed and plotted throughout the text.

The networks trained on MNIST and CIFAR-10 were implemented using Keras and Tensorflow, and trained for a fixed number of epochs with the ADAM optimizer.

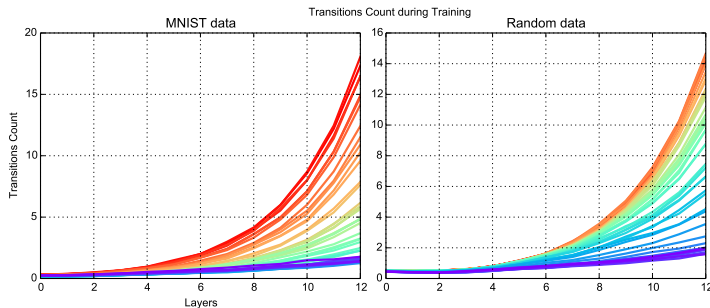

Figure 11: An identical plot to Figure 8 but for transition count.

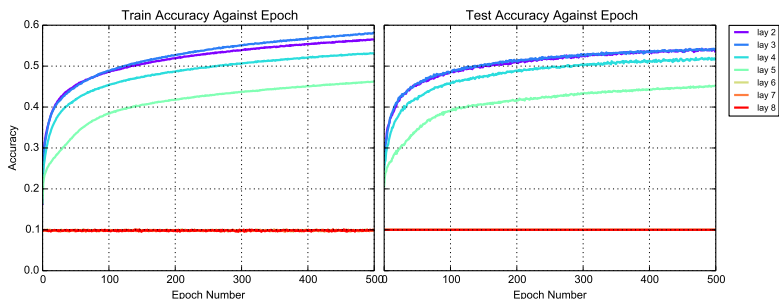

Figure 12: We repeat the experiment in Figure 6 for a convolutional network trained on CIFAR-10. The network has eight convolutional hidden layers, with three by three filters and $64$ filters in each layer, all with ReLU activations. The final layer is a fully connected softmax, and is trained in addition to the single convolutional layer being trained. The results again support greater expressive power with remaining depth. Note the final three convolutional layers failed to effectively train, and performed at chance level.

We also have preliminary experimental results on Convolutional Networks. To try and make the comparisons fair, we implemented a fully convolutional network (no fully connected layers except for the last layer).

We also include the plot showing the effect of training on number of transitions for interpolated MNIST and interpolated random points.

