# Peer review of "On the Expressive Power of Deep Neural Networks"

_ICLR 2017 — rejected_

[Public Comment · Tara N Sainath · 07 Nov 2016]
**ICLR Paper Format**

Dear Authors,

Please resubmit your paper in the ICLR 2017 format with the correct font type for your submission to be considered. Thank you!

[Official Review · AnonReviewer2 · rating 5 · confidence 3 · 15 Dec 2016]
**Interesting ideas on the trajectory lengths, the motivations and the conclusion of the study are not clear**

Summary of the paper:

Authors study in this paper quantities related to the expressivity of neural networks.The analysis is done for a random network. authors define the ‘trajectory length’ of a one dimensional trajectory as the length of the trajectory as the points (in a m- dimensional space) are embedded by layers of the network. They provide growth factors as function of hidden units k, and number of layers d.  the growth factor is exponential in the number of layers. Authors relates this trajectory length to authors quantities : ‘transitions’,’activation patterns ’ and ‘Dichotomies’. 
As a consequence of this study authors suggest that training only  earlier layers in the network  leads higher accuracy then just training later layers. Experiments are presented on MNIST and CIFAR10.

Clarity:

The  paper is a little hard to follow, since  the motivations are not clear in the introduction and the definitions across the paper are not clear. 

Novelty:

Studying the trajectory length as function of transforming the data by a multilayer network is   new and interesting idea. The relation to transition numbers is in term of the growth factor, and not as a quantity to quantity relationship. Hence it is hard to understand what are the implications.

Significance:

The geometry of the input set (of dimension m)  shows up only weakly in the activation patterns analysis.  The trajectory study should tell us how the network organizes the input set. As observed in the experiments the network becomes contractive/selective as we train the network. It would be interesting to study those phenomenas using this trajectory length , as a measure for disentangling nuisance factors ( such as invariances etc.). In the supervised setting the network need not to be contractive every where , so it needs to be selective to the class label, a  theoretical study of the selectivity and contraction using the trajectory length would be more appealing.

Detailed comments:

Theorem 1:

- As raised by reviewer one the definition of a one dimensional input trajectory is missing. 
- What does theorem 1 tells us about the design and the architecture to use in neural networks as promised in the introduction is not clear. The connection to transitions in Theorem 2 is rather weak. 

Theorem 2:

- in the proof of theorem 2 it not clear what is meant by T and t. Notations are confusing, the expectation is taken with respect to which weight: is it W_{d+1} or (W_{d+1} and W_{d})? I understand you don't want to overload notation but maybe E_{d+1} can help keeping track. I don't see how the recursion is applied if T and t in it, have different definitions. seems T_{d+1} for you is a random variable and t_{d} is fixed. Are you fixing W_d and then looking at W_{d+1} as  random?

- In the same proof:  the recursion  is for d>1  ? your analysis is for W \in R^{k\times k}, you don't not study the W \in \mathbb{R}^{k\times m}. In this case you can not assume assume that |z^(0)|=1.

- should d=1, be analyzed alone to know how it scales with m?

Theorem 4 in main text:

- Is the proof missing? or Theorem 4 in the main text is Theorem 6 in the appendix?

Figures 8 and 9:

- the trajectory length reduction in the training isn't that just the network becoming contractive to enable mapping the training points to the labels? See for instance  on contraction in deep networks

[Author Response · Maithra Raghu · 17 Dec 2016]
**Response to Reviewer2**

Thank you for the review! We will take the comments into account and endeavour to make the text even clearer.

A quick comment about motivation: our goal in this work improve interpretability in deep neural networks through a better understanding of neural network expressivity. In particular, we look at different "diagnostics" (transitions/activation patterns/dichotomies) for measuring the expressiveness of different neural network architectures, and their practical consequences. The surprising fact that three natural measures of expressiveness are related by *direct* proportion (see below) to trajectory length suggests consequences on remaining depth (earlier parameters are more important to fit the final function) and a trade off between expressivity and stability during training due to initialization choices.  

Responses inline to other specific comments below:

Trajectory Length: We will add this as a definition before Theorem 1. We take a 1-d trajectory to be a 1-d curve in the high dimensional space, and we measure the length  --

[Official Review · AnonReviewer1 · rating 6 · confidence 5 · 19 Dec 2016]
**Review of ``ON THE EXPRESSIVE POWER OF DEEP NEURAL NETWORKS''**

SUMMARY 
This paper studies the expressive power of deep neural networks under various related measures of expressivity. 
It discusses how these measures relate to the `trajectory length', which is shown to depend exponentially on the depth of the network, in expectation (at least experimentally, at an intuitive level, or theoretically under certain assumptions). 
The paper also emphasises the importance of the weights in the earlier layers of the network, as these have a larger influence on the represented classes of functions, and demonstrates this in an experimental setting. 

PROS 
The paper further advances on topics related to the expressive power of feedforward neural networks with piecewise linear activation functions, in particular elaborating on the relations between various points of view. 

CONS 
The paper further advances and elaborates on interesting topics, but to my appraisal it does not contribute significantly new aspects to the discussion. 

COMMENTS
- The paper is a bit long (especially the appendix) and seems to have been written a bit in a rush. 
Overall the main points are presented clearly, but the results and conclusions could be clearer about the assumptions / experimental vs theoretical nature. 
The connection to previous works could also be clearer. 

- On page 2 one finds the statement ``Furthermore, architectures are often compared via ‘hardcoded’ weight values -- a specific function that can be represented efficiently by one architecture is shown to only be inefficiently approximated by another.'' 

This is partially true, but it neglects important parts of the discussion conducted in the cited papers. 
In particular, the paper [Montufar, Pascanu, Cho, Bengio 2014] discusses not one hard coded function, but classes of functions with a given number of linear regions. 
That paper shows that deep networks generically* produce functions with at least a given number of linear regions, while shallow networks never do. 
* Generically meaning that, after fixing the number of parameters, any function represented by the network, for parameter values form an open, positive -measure, neighbourhood, belongs to the class of functions which have at least a certain number of linear regions. 
In particular, such statements can be directly interpreted in terms of networks with random weights. 

- One of the measures for expressivity discussed in the present paper is the number of Dichotomies. In statistical learning theory, this notion is used to define the VC-dimension. In that context, a high value is associated with a high statistical complexity, meaning that picking a good hypothesis requires more data. 

- On page 2 one finds the statement ``We discover and prove the underlying reason for this – all three measures are directly proportional to a fourth quantity, trajectory length.'' 
The expected trajectory length increasing exponentially with depth can be interpreted as the increase (or decrease) in the scale by a composition of the form a*...*a x, which scales the inputs by a^d. Such a scaling by itself certainly is not an underlying cause for an increase in the number of dichotomies or activation patterns or transitions. Here it seems that at least the assumptions on the considered types of trajectories also play an important role. 
This is probably related to another observation from page 4: ``if the variance of the bias is comparatively too large... then we no longer see exponential growth.''

OTHER SPECIFIC COMMENTS 
In Theorem 1 
- Here it would be good to be more specific about ``random neural network'', i.e., fixed connectivity structure with random weights, and also about the kind of one-dimensional trajectory, i.e., finite in length, closed, differentiable almost everywhere, etc. 

- The notation ``g \geq O(f)'' used in the theorem reads literally as |g| \geq \leq k |f| for some k>0, for large enough arguments. It could also be read as g being not smaller than some function that is bounded above by f, which holds for instance whenever g\geq 0. 
For expressing asymptotic lower bounds one can use the notation \Omega (see

[Official Review · AnonReviewer3 · rating 3 · confidence 3 · 23 Dec 2016]
**Not clear. The approach and methodology are not explained.**

This paper presents a theoretical and empirical approach to the problem of understanding the expressivity of deep networks.

Random networks (deep networks with random Gaussian weights, hard tanh or ReLU activation) are studied according to several criterions: number of neutron transitions, activation patterns, dichotomies and trajectory length.

There doesn't seem to be a solid justification for why the newly introduced measures of expressivity really measure expressivity.
For instance the trajectory length seems a very discutable measure of expressivity. The only justification given for why it should be a good measure of expressivity is proportionality with other measures of expressivity in the specific case of random networks.

The paper is too obscure and too long. The work may have some interesting ideas but it does not seem to be properly replaced in context.

Some findings seem trivial.

detailed comments

p2 

"Much of the work examining achievable functions relies on unrealistic architectural assumptions such as layers being exponentially wide"

I don’t think so. In "Deep Belief Networks are Compact Universal Approximators" by Leroux et al., proof is given that deep but narrow feed-forward neural networks with sigmoidal units can represent any Boolean expression i.e. A neural network with 2n−1 + 1 layers of n units (with n the number of input neutron).

“Comparing architectures in such a fashion limits the generality of the conclusions”

To my knowledge much of the previous work has focused on mathematical proof, and has led to very general conclusions on the representative power of deep networks (one example being Leroux et al again).

It is much harder to generalise the approach you propose, based on random networks which are not used in practice.

“[we study] a family of networks arising in practice: the behaviour of networks after random initialisation”

These networks arise in practice as an intermediate step that is not used to perform computations; this means that the representative power of such intermediate networks is a priori irrelevant. You would need to justify why it is not.

“results on random networks provide natural baselines to compare trained networks with”

random networks are not “natural” for the study of expressivity of deep networks. It is not clear how the representative power of random networks (what kind of random networks seems an important question here) is linked to the representative power of (i) of the whole class of networks or (ii) the class of networks after training. Those two classes of networks are the ones we would a priori care about and you would need to justify why the study of random networks helps in understanding either (i) or (ii).

p5

“As FW is a random neural network […] it would suggest that points far enough away from each other would have independent signs, i.e. a direct proportionality between the length of z(n)(t) and the number of times it crosses the decision boundary.”

As you say, it seems that proportionality of the two measures depends on the network being random. This seems to invalidate generalisation to other networks, i.e. if the networks are not random, one would assume that path lengths are not proportional.

p6

the expressivity w.r.t. remaining depth seems a trivial concerns, completely equivalent to the expressivity w.r.t. depth. This makes the remark in figure 5 that the number of achievable dichotomies only depends *only* on the number of layers above the layer swept seem trivial

p7

in figure 6 a network width of 100 for MNIST seems much too small. Accordingly performance is very poor and it is difficult to generalise the results to relevant situations.

[Final Decision · Program Chairs · 06 Feb 2017]
**ICLR committee final decision**

While the reviewers saw some value in your contribution, there were also serious issues, so the paper does not reach the acceptance threshold.